# Multidomain architecture of estrogen receptor reveals interfacial cross-talk between its DNA-binding and ligand-binding domains

Wei Huang[1], Yi Peng[1], Janna Kiselar[1], Xuan Zhao[2], Aljawharah Albaqami[2], Daniel Mendez[2], Yinghua Chen[3], Srinivas Chakravarthy[4], Sayan Gupta[5], Corie Ralston[5], Hung-Ying Kao[2], Mark R. Chance[1] & Sichun Yang [1]

Human estrogen receptor alpha (hERα) is a hormone-responsive nuclear receptor (NR) involved in cell growth and survival that contains both a DNA-binding domain (DBD) and a ligand-binding domain (LBD). Functionally relevant inter-domain interactions between the DBD and LBD have been observed in several other NRs, but for hERα, the detailed structural architecture of the complex is unknown. By utilizing integrated complementary techniques of small-angle X-ray scattering, hydroxyl radical protein footprinting and computational modeling, here we report an asymmetric L-shaped "boot" structure of the multidomain hERα and identify the specific sites on each domain at the domain interface involved in DBD–LBD interactions. We demonstrate the functional role of the proposed DBD–LBD domain interface through site-specific mutagenesis altering the hERα interfacial structure and allosteric signaling. The L-shaped structure of hERα is a distinctive DBD–LBD organization of NR complexes and more importantly, reveals a signaling mechanism mediated by inter-domain crosstalk that regulates this receptor's allosteric function.

[1] Center for Proteomics and Department of Nutrition, Case Western Reserve University, 2109 Adelbert Rd, Cleveland, OH 44106-4988, USA. [2] Department of Biochemistry, Case Western Reserve University, 10900 Euclid Ave, Cleveland, OH 44106, USA. [3] PEPCC Facility, Case Western Reserve University, 10900 Euclid Ave, Cleveland, OH 44106, USA. [4] BioCAT-18ID, Advanced Photon Source, Argonne National Laboratory, 9700 S. Cass Avenue, Argonne, IL 60439, USA. [5] Molecular Biophysics and Integrated Bioimaging, Lawrence Berkeley National Laboratory, Berkeley, CA 94720, USA. Correspondence and requests for materials should be addressed to S.Y. (email: sichun.yang@case.edu)

The human estrogen receptor alpha (hERα) is a hormone-responsive nuclear receptor (NR) involved in physiological processes such as cell growth, survival, and cancer metastasis[1,2]. Activated by its cognate hormone estradiol, hERα functions as a homodimer and regulates transcription by binding specific DNA sequences in target genes[3]. Like other NRs, it contains a highly conserved DNA-binding domain (DBD) and a C-terminal 12-helical ligand-binding domain (LBD)[4] (Fig. 1a). Several other NRs, including PPARγ-RXRα[5], VDR–RXRα[6], RARα-RXRα[7], HNF-4α homodimer[8], RXRα-LXRβ[9], USP/EcP[10], and more recently, RARβ-RXRα[11], have been characterized with respect to their physical interactions and the allosteric communication between the DBD and LBD. These distinct DBD–LBD interactions mediate allosteric signal transduction in the function of the different NRs. For hERα, however, a lack of information on the DBD–LBD interaction has made it impossible to dissect the inner-workings of receptor activation critical for hormonal signaling.

Current understanding of the mechanistic action of hERα has mostly relied on analyses of the individual DBDs or LBDs. For example, the crystal structure of the DBD homodimer shows that the DBD binds a consensus palindromic DNA duplex known as estrogen response element (ERE) (Fig. 1b), while the LBD homodimer in complex with estradiol and coactivator TIF2 peptides shows that the hormone is capped in place by its C-terminal helix H12[12–16] (Fig. 1c). For other NRs, solution-phase biophysical techniques, including small-angle X-ray scattering (SAXS), fluorescence resonance energy transfer, and H/D exchange, have provided various levels of structural information for complexes including RXRα–RARα and VDR–RXRα[6,7]. One recent structural study of hERα using cryo-electron microscopy (EM) showed hERα in complex with coactivators within a large transcription complex[17,18], although this EM study at a 22-Å resolution did not provide a detailed picture of the DBD–LBD architecture[17]. Our own computational docking studies of individual hERα domains have revealed a variety of likely hERα structures[19], but the selection of reliable conformations has remained speculative due to the lack of experimental support, pointing to the need for a molecular understanding of the hERα complex and its domain interactions.

To investigate how the different domains within the hERα interact with each other, we conducted multiple highly complementary, in-solution biophysical studies using a recombinant, active form of the hERα protein. To identify specific sites of domain interaction, we first used a hydroxyl radical-based protein footprinting, where hydroxyl radicals (generated by radiolysis) react with solvent accessible side chains and the sites and rates of oxidation are monitored by quantitative mass spectrometry (MS) [20]. The resultant data provide a measure of surface accessibility of individual residue side chains[21,22]. We identified two separate clusters of hydrophobic residues that are on the surface of each of the domain-dimers in the isolated states (as seen from their

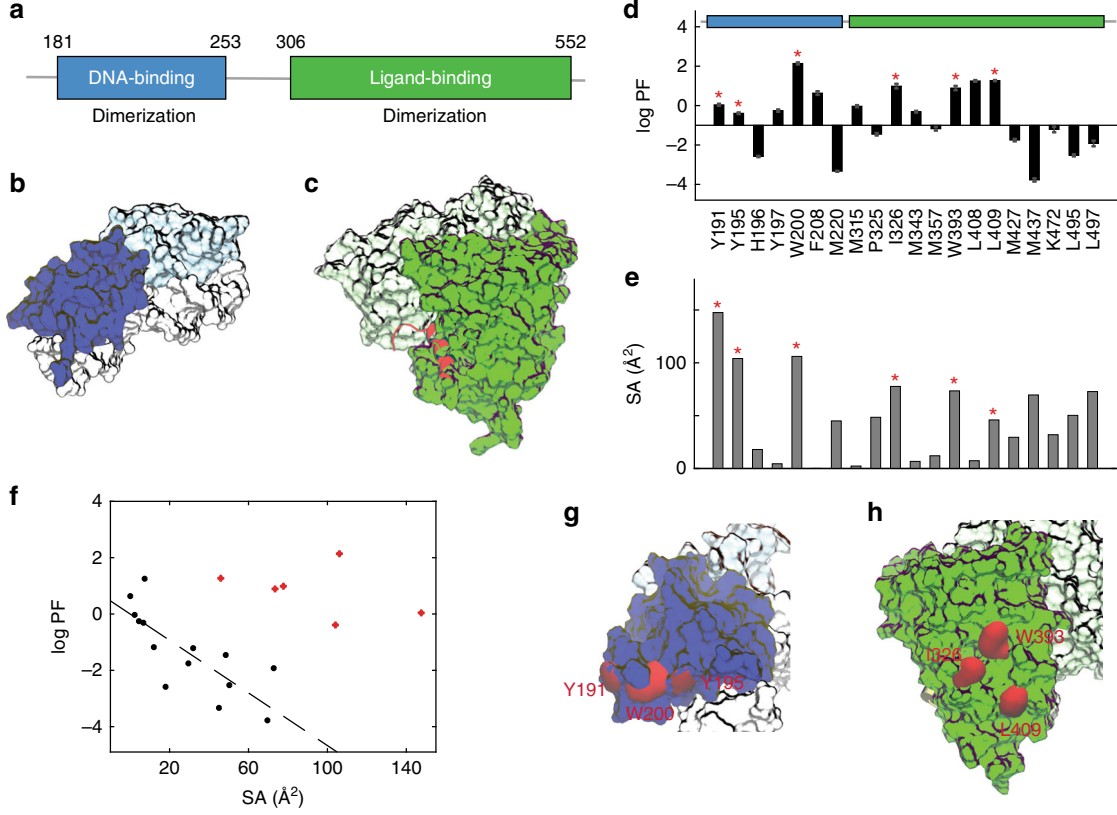

**Fig. 1** Contact residues between the DBD and LBD identified by footprinting. **a** Structural domains of hERα. Human ERα contains a DNA-binding domain (DBD; blue), a ligand-binding domain (LBD; green), and functions as a homodimer. **b**, **c** The crystal structures of DBD dimer (**b** light/dark blue) in complex with ERE–DNA (gray) (1HCQ.pdb), and of LBD dimer (**c** light/dark green) in complex with estradiol and a coactivator TIF2 peptide (1GWR.pdb). The C-terminal helix H12 of the LBD is highlighted (red). **d** Hydroxyl radical footprinting of hERα. High logPF values of six residues (red asterisks) indicate their involvement in domain contacts. Duplicates were performed and standard deviations were indicated. **e** Solvent accessibility surface area (SA) values of residue side chains calculated from the crystal structure of individual domains. **f** Correlation between logPF and SA values. Differentiation of the six contact residues (red dots) is shown from the rest of 14 residues (black dots). The latter have a Pearson's correlation coefficient −0.77 (p-value = 0.001). **g**, **h** Structural mapping of contact residues. Contact residues (red) are Y191/Y195/W200 on the surface of the DBD (blue blobs) and I326/W393/L409 on the LBD (green blobs)

crystal structures), but are much less solvent accessible in the examination of the complex. To investigate the overall domain arrangement, we next acquired SAXS data of the complex in solution, thereby enabling the complementary integration of SAXS data with residue-level information from footprinting to elucidate the hERα DBD–LBD architecture and identify a network of residue–residue interactions at the DBD–LDB interface. We validated our findings by employing site-directed mutagenesis followed by functional transcriptional and DNA-binding studies. We further investigated if the mutations can influence the structural stability across the interface to a distal domain, using intrinsic tryptophan fluorescence. The observation that interfacial mutations alter the hERα structure and function establishes the existence of a previously uncharacterized interaction between the DBD and LBD, and further demonstrates the functional relevance of the DBD–LBD interface. Notably, the L-shaped boot structure of the receptor represents a distinctive architecture of the DBD–LBD spatial organization that can be used to interpret the functional relation of clinical mutations and provides a structural basis for developing small molecules by disrupting the cross-talk at the DBD–LBD interface to regulate receptor function.

## Results

**Specific residue sites involved in DBD–LBD contact.** We generated a multidomain segment of the hERα containing both DBD and LBD (residues E181–P552, referred to as hERα$^{CDE}$ in Fig. 1a), and purified hERα$^{CDE}$ proteins using size exclusion chromatography (Supplementary Fig. 1), in the presence of the receptor's ligands: estradiol (E2), an ERE-containing DNA duplex (ERE–DNA; 5′-TAGGTACACGTGACCTGCG-3′ and 5′-CGCAGGTCACTGTGACCTA-3′), and a coactivator TIF2 peptide (KENALLRYLLDKDD), as in previous crystallographic studies (Fig. 1b, c). We hereafter refer to this as hERα$^{complex}$. Purified hERα$^{complex}$ samples at micromolar concentrations were exposed to a focused synchrotron X-ray white beam, the sites of oxidation were detected by tandem MS (MS/MS), and the extent of modification was quantified as a function of X-ray dose (see examples in Supplementary Fig. 2). These curves were fit to an exponential decay function and the measured rate constants ($k_{fp}$) were divided by a measure of their intrinsic reactivity with hydroxyl radicals ($k_R$), thereby providing a residue-level protection factor (PF = $k_R/k_{fp}$)[22,23]. The log of the PF values provides an accurate surface topology map, where high logPF values reflect more protection for solvent and lower logPF values reflect greater solvent exposure for the set of 20-probed residues across the receptor (Fig. 1d), as reflected by the correlation between logPF and solvent accessibility of the hERα residues (Fig. 1f).

We compared the measured logPF values for the 20 residues (Fig. 1d) with their solvent accessible surface areas (SA) extracted from the crystal structures of their individual domain-dimers (Fig. 1e). Our expectation was that residues involved in the interface are expected to have large SA values as in the isolated domains, but also have high logPF values (i.e., be solvent-protected) in the complex. For the ten most solvent-exposed residues for the isolated domain structures (with SA values ranging from 45.9 to 147.6 Å$^2$), only six exhibit high logPF values (ranging from −0.39 to 2.13) for the hERα$^{complex}$ (red stars, Fig. 1d). In contrast, other highly protected residues among the 20, like M343, Y197, M315, F208, and L408 (logPF = −0.31, −0.26, −0.03, 0.64, and 1.25, respectively) have SA values from crystallography of less than 15 Å$^2$ (Supplementary Table 1), indicating that they are protected for the individual domains as well as the observed complex. The differentiation of the six from the remaining 14 residues is also demonstrated by examining the relationship of logPF and SA for the 14 residues. The Pearson's

correlation coefficient, $p = -0.77$ ($p$-value = 0.001) (black dots in Fig. 1f), suggests that the observed logPF values for these residues are consistent with the isolated domain structures, while the six suggested contact residues (red) show a significant departure from the fit line and an overall inconsistency with the individual domain structures. Other residues highly exposed for the individual domains (e.g., M220 and M437) remain fully solvent-exposed in the complex (logPF = −3.33 and −3.78, respectively), ruling them out as candidates for the interface as well.

Of the six candidate residues, W200 is fully exposed in the crystal structure of the DBDs (SA = 106.1 Å$^2$), while it has the highest logPF value (logPF = 2.14) for hERα$^{complex}$, a strong evidence that it is highly buried within DBD–LBD contacts. Similarly, I326 and W393 experience a notable change in solvent accessibility with relatively high logPF values in the complex (logPF = 0.99 and 0.89, respectively), but are fully solvent exposed on the individual domain surface (SA = 77.7 and 73.4 Å$^2$, respectively). From a quantitative perspective, we can use the relationship of Fig. 1f to infer the SA values for the six putative interfacial residues in hERα$^{complex}$, suggesting the six residues experience SA decrease ranging from 53 Å$^2$ (L409) to 157 Å$^2$ (Y191) for the complex, compared to their SA values in the individual domains (Supplementary Table 1). Consistent with these residues participating in the architecture of the interface, these residues define two separate hydrophobic clusters in the hERα$^{complex}$. One hydrophobic cluster is formed among residues Y191, Y195, and W200 on the DBD surface (Fig. 1g), and the other is on the LBD including I326, W393, and L409 (Fig. 1h). Overall, the observed high logPF values of these hydrophobic residues coupled with the fact that they are solvent exposed on the domains' surfaces and their clustering as patches on the LDB and DBD all point to their involvement in DBD–LBD interactions. H, the mode of interaction between the domains is not clear from the footprinting data alone.

**Overall architecture of the homodimeric hERα complex.** To investigate the actual domain arrangement between the DBD and LBD, we next acquired small-angle X-ray scattering (SAXS) data of the hERα$^{complex}$ in solution. SAXS provides structural information about spatial organization of the domains[24–26]. By using the elution peak of purified hERα$^{CDE}$ proteins in the presence of the receptor's ligands, an "on-the-fly" SAXS data acquisition was achieved for the hERα$^{complex}$ via a chromatography-coupled setup (Supplementary Fig. 3). This approach mitigates potential protein aggregates and contamination from excess ligands[27]. The ab initio SAXS reconstruction alone has been successfully applied to visualize the overall shapes of several NR complexes[7]. By combining domain-arrangement data from SAXS with protector factors of surface residues from footprinting, a detailed and reliable picture for hERα$^{complex}$ can be constructed. As our laboratory and others have successfully demonstrated[28–34], this experiment-directed integrative approach has proven valuable in overcoming the limitations of individual techniques, in this case suitable for elucidating the hERα structure.

To determine the three-dimensional organization and structure of hERα, we scored computational docking models against experimental SAXS (domain arrangement) and footprinting (contact site) data via an in-house multi-technique iSPOT platform[29,30]. Computationally docked structures were generated in two steps of (1) rigid-body docking and coarse-grained sampling[19,29], and (2) atomic-level simulations with distance restraints between the two clusters Y191/Y195/W200 and I326/W393/L409 (Methods). The former sampling was among rotational and translational motions between the domains (i.e.,

five initial poses per rotation and a range of 0–50 Å for translation) to achieve an extensive search (Supplementary Fig. 4), while the latter atomic-level simulations were distance restrained linearly from 2 to 10 Å between the centers of mass of the two clusters to concentrate on local sampling. The goodness of fit of each structural candidate against our experimental SAXS and footprinting data was evaluated via two scoring functions $\chi^2$ and $\varphi^2$. The unit-less $\chi^2$ is defined to measure the difference between the theoretical and experimental SAXS profile by[28],

$$\chi^2 = \frac{1}{N_q} \sum_q \frac{\left\{ \log I_{cal}(q) - \log I_{exp}(q) \right\}^2}{\sigma^2(q)}, \tag{1}$$

where $I_{cal}$ is calculated using fast-SAXS-pro[35] and $N_q$ is the number of scattering $q$ points recorded in experimental $I_{exp}$ (with its measurement error of $\sigma(q)$). Similarly, the $\varphi^2$ is the goodness of fit between experimental and theoretical footprinting data by[30],

$$\varphi^2 = \frac{1}{N_s} \sum_s \frac{\left\{ \log PF_{cal}(s) - \log PF_{exp}(s) \right\}^2}{\delta^2(s)}, \tag{2}$$

where $\log PF_{cal}$ is the predicted logPF value of each site based on its corresponding SA value, using the linear regression between experimental $\log PF_{exp}$ values (with an error $\delta(s)$ at each site ($s$)) and SA values of $N_s$ sites for each docked structure. The precise evaluation by $\chi^2$ and $\varphi^2$ enables the differentiation among all docked structural candidates (Supplementary Fig. 5), i.e., a lower $\chi^2$ and $\varphi^2$ value indicates a better fit with experimental scattering and footprinting data at the domain arrangement and local contact-site level (Fig. 2a), respectively. This permits the identification of the best-fit ensemble structures (Fig. 2b, c) that fit both scattering and footprinting data simultaneously. The latter are within 3 Å of Cα-RMSD from the former best-fit structure (Supplementary Fig. 6).

Overall, the multidomain hERα exhibits an asymmetric organization that resembles an asymmetric L-shaped "boot" (Fig. 2b, c). One of the LBDs leans against its own DBD (at the "tongue" position of the boot) with its interacting ERE–DNA below (the "outsole" position), while the LBD dimer (light/dark green for each monomer) form the boot "shaft" and lie perpendicular to the plane of the DBD dimer (light/dark blue) and the ERE–DNA (gray). Based on the observed ensemble structures (Fig. 2b), a large Cα-atom RMS fluctuation (Cα-RMSF) is observed at the domain-connecting region ranging from 5 to 10 Å, although most tertiary domains and footprinting-detected residues have a Cα-RMSF value of around 2–4 Å (Supplementary Fig. 7). This large fluctuation at the domain-connecting region makes its placement with respect to the DBD and LBD less certain. We note that it is possible to have large flexibility with the loop region (e.g., in a Cα-RMSF range of 10–30 Å) (Supplementary Fig. 8a), although its contribution to overall scattering is relatively small with the overall $\chi^2 = 1.2 \pm 0.1$ (Supplementary Fig. 8b), where the large structural fluctuation in the loop (with a RMSD value up to 23 Å) contributes <10% to the overall $\chi^2$ value of the entire complex. While it appears that the LBD interacts with DBD within the same polypeptide chain, we could not exclude the possibility that the DBD and LBD are each from a different polypeptide chain (e.g., domain swapping), despite the strong binding between the LBD monomers within its dimeric complex as well as strong interactions of the DBD dimer with the ERE–DNA duplex. It also appears that modest conformational deformation occurs in the LBD–DBD interface upon complex formation, compared to the crystal structures of individual

domains, especially for the DBD contact residues (Supplementary Fig. 9).

The goodness of fit of the hERα ensemble structures is evidenced by the agreement between experimental and calculated scattering and footprinting data. Comparison of the calculated and experimental SAXS profile yields $\chi^2 = 3.0 \pm 0.1$ (using fast-SAXS-pro[35]), and an even better score $\chi^2 = 1.2 \pm 0.1$ (using CRYSOL[36]) (Fig. 2d). Based on the same ensemble structures, the average SAs are linearly correlated with their corresponding measured logPF values at a correlation coefficient −0.95 ($p$-value = 0.002) for the observed sites (Fig. 2e). Whereas an absolute measure of model resolution is not apparent from the goodness of fit, the Cα-RMSD of the hERα ensemble structures provides a reliable description (using one of its own best-fit structures as a reference), with regard to the final models' accuracy (Supplementary Fig. 6). In addition, our prior theoretical study using simulated scattering and footprinting data of the structurally known HNF-4α homodimer, similar to hERα in size, predicted a very close RMSD value of 4.2 Å (excluding domain-connecting loop regions), compared to its solved crystal structure[29]. Notably, the observed hERα domain arrangement within its boot-like architecture fits qualitatively into the EM map at a 22-Å resolution (EM Data Bank EMD-8832) of the full-length hERα (Supplementary Fig. 10)[17]. Taken together, this consistency nicely affirms the positioning of the observed asymmetric domain arrangement.

**The DBD–LBD interface and its functional importance.** The observed domain arrangement and mode of interaction between the DBD and LBD warranted a residue–residue contact analysis at the DBD–LBD interface. By examining the molecular surfaces that interact with one another using the contact of structural units (CSU) approach[37], a well-formed interface was observed between the DBD and LBD (Fig. 3a). The interface is mainly composed of hydrophobic contacts (Fig. 3b), between the DBD's residues, including Y191, Y195, G198, V199, and W200 right before the first helix of the DBD and the LBD's eight contact residues, including I326, Y328, W393, E397, L403, P406, N407, and L409. Using the conventional hERα-LBD nomenclature[15] (Supplementary Fig. 11), we found that the LBD contact residues are positioned in the two-strand region between helices H5 and H6 (L403/P406/N407/L409), the end of helix H5 (W393), and the loop region between helices H1 and H3 (I326/Y328). A close-up view shows a modest-sized pocket on the LBD surface in contact with the DBD (Supplementary Fig. 12), where residue W200 becomes buried and protected from solvent exposure at the interface, consistent with its lack of radiolytic labeling (Supplementary Fig. 2) and its high protection factor (Fig. 1d).

The functional importance of the DBD–LBD interface was first explored by introducing mutations at this domain–domain junction and analyzing their effects on transcriptional activity. The introduction of point mutations on the LBD, namely, I326A, Y328A, P406A, and L409A (depicted in black in Fig. 4a), considerably reduced the transcriptional activity when compared to the wild-type protein (Fig. 4b). We further generated Gal4-DBD/hERα-LBD fusion proteins for both wild-type and mutant hERα-LBD, and found that these mutant fusion proteins possess comparable E2-induced reporter activity to that of the wild-type hERα-LBD (Fig. 4c). These results argue against the possibility that these hERα-LBD mutants loss their hormone or coactivator-binding activities, further confirming the functional significance of these interfacial residues in mediating the receptor's transcriptional function through the LBD–DBD interface. Interestingly, mutations at these sites, such as I326, P406, and L409, have been identified in cancer patient samples[38–40]. While their oncogenic

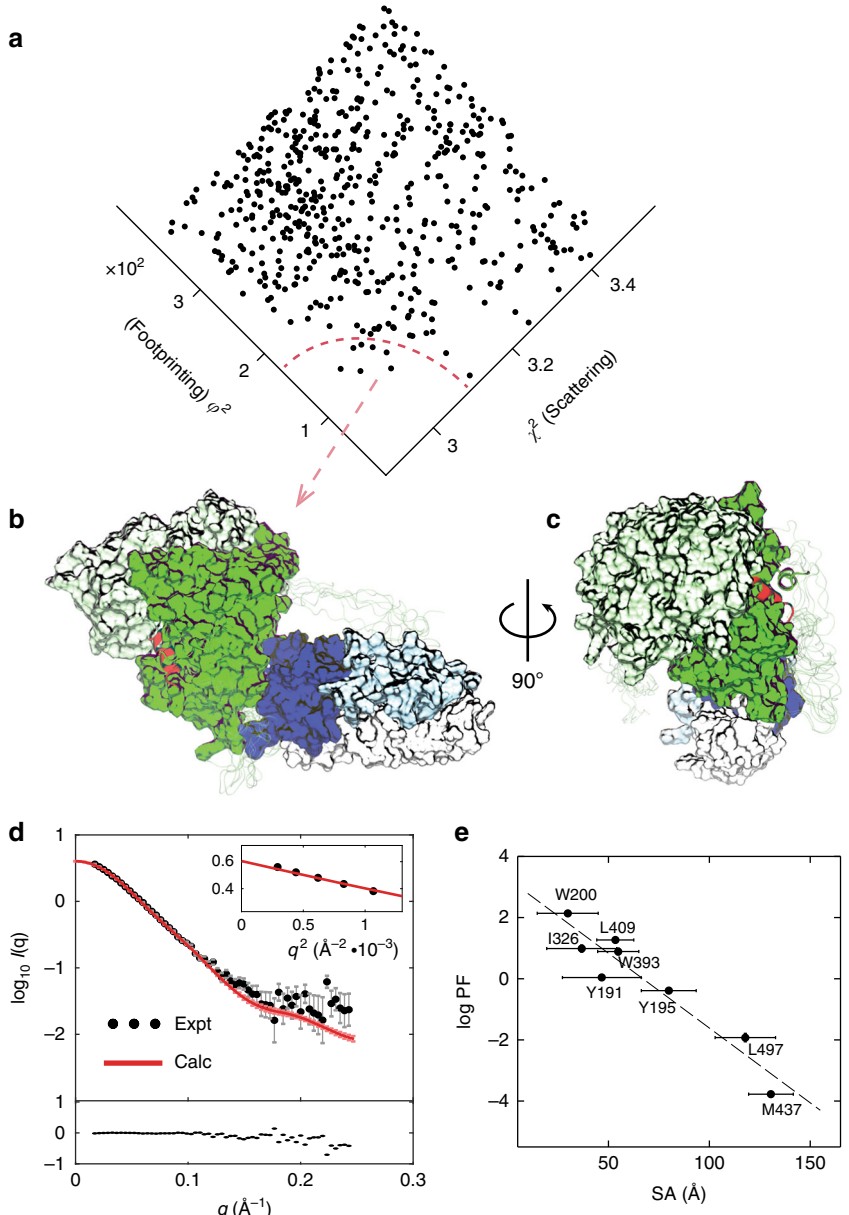

**Fig. 2** Overall architecture of the hERα homodimer revealed by data integration. **a** Fitting against experimental data. The fit of computationally generated conformations (dot) is simultaneously assessed against hydroxyl radical protein footprinting ($\varphi^2$) and small-angle X-ray scattering ($\chi^2$). Lower $\chi^2$ and $\varphi^2$ values are better in fitting. The best-fit ensemble structures lie at the bottom corner of the fit plot, below the red dashed line. **b** Ensemble of best-fit hERα structures. It contains both LBD monomers (light/dark green) and DBD monomers (light/dark blue). The C-terminal helix H12 of the LBD is in red, and ERE–DNA is in gray. The LBD–DBD connecting loops are shown as light green ribbons. The structure models (within the red circle) are within 3 Å Cα-RMSD of the best-fit structure. **c** A rotated view of the best-fit hERα structures. **d** Goodness of fit to measured SAXS data. Theoretical SAXS data were the ensemble average of the set of hERα structures above. The scattering intensity, $\log_{10} I(q)$, is plotted as a function of the scattering angle ($q$). The goodness of fit $\chi^2 = 1.2$. Inserted is the Guinier plot with a linear fit, yielding the radius of gyration $R_g = 38.0 \pm 0.3$ Å. The bottom graph shows residuals from subtraction between calculated and experimental profiles. A total of six scattering images were used and standard deviations were indicated. **e** Goodness of fit to footprinting data. Measured footprinting protection factors (logPF) are plotted against average accessible surface areas (SA) derived from the ensemble structures. Linear correlation coefficient is $\rho = -0.95$. A total of seven structures were used for ensemble calculations and standard deviations were indicated

relevance remains to be established, alteration of the DBD–LBD interface by these mutations on reporter activity may provide a functional link to their ability to regulate transcription.

Notably, the substitution of a non-charged residue Y191 with a charged histidine resulted in increased transcriptional activity (Fig. 3c). From a structural perspective, the Y191H mutation likely resulted in a local energetic stabilization due to a stronger interaction with its oppositely charged neighbor residue, E397,

across the interface. To further assess the impact of this substitution on the receptor's ability to bind ERE–DNA, a biochemical DNA-binding assay was performed by detecting fluorescence anisotropy of 6-carboxyfluorescein-labeled ERE–DNA in the presence of increasing concentrations of the E2-bound hERα$^{CDE}$. The assay was applied to WT, W393A, and Y191H conditions, respectively, to obtain their DNA-binding affinity $K_d$ values (Methods). As a negative control, W393A had

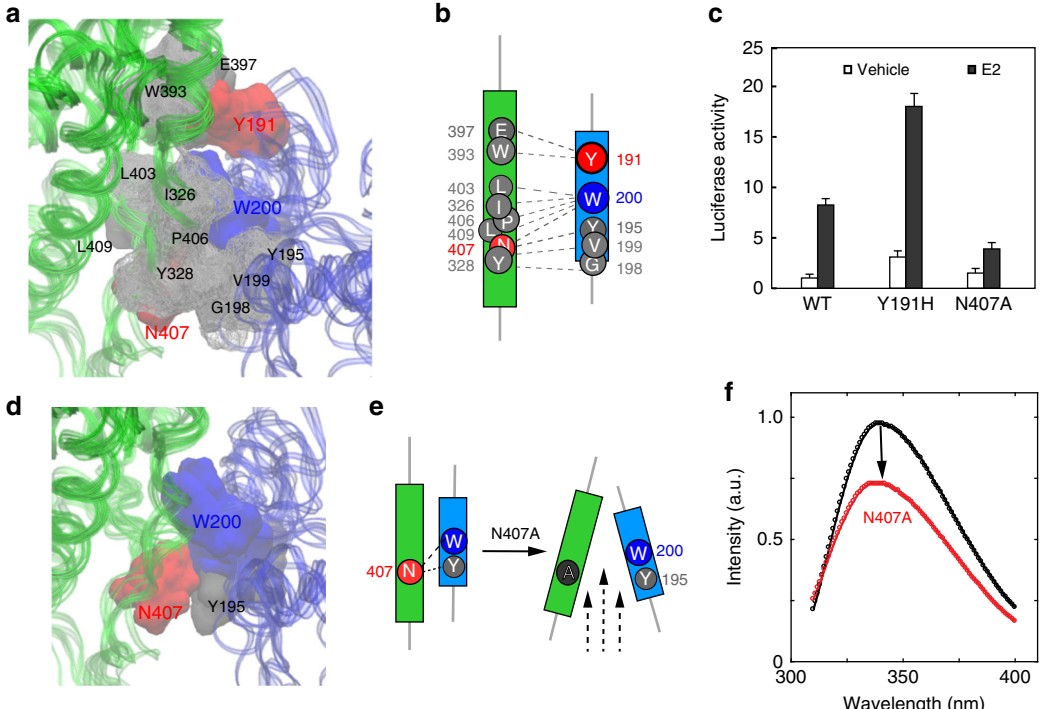

**Fig. 3** The DBD–LBD interface and its functional relevance. **a** Close-up view of the DBD–LBD interface. Highlighted are contact residues with mutation sites (red) and the fluorescence site at W200 (blue). **b** Cartoon of interfacial residues. Dashed lines indicate a probability >75% of making a residue contact within the structure ensemble. **c** Effect of interfacial mutations on ER transcription activity. The Y191H mutation increases the transcription luciferase activity of the receptor, while N407A reduces the activity. Triplicates were carried out and standard deviations were indicated. **d** Tryptophan fluorescence site at W200. Shown are interactions between hydrophobic residues N407 (red circle), Y195 (gray circle), and W200 (blue circle) at the interface. **e** A schematic representation of tryptophan surroundings upon mutation. Illustrated are possible structural changes near W200 before and after mutation. **f** Quenching of tryptophan fluorescence. Emission fluorescence intensity is reduced in mutant N407A. A protein concentration of 0.1 mg/ml was used before and after mutation. Excitation at 295 nm

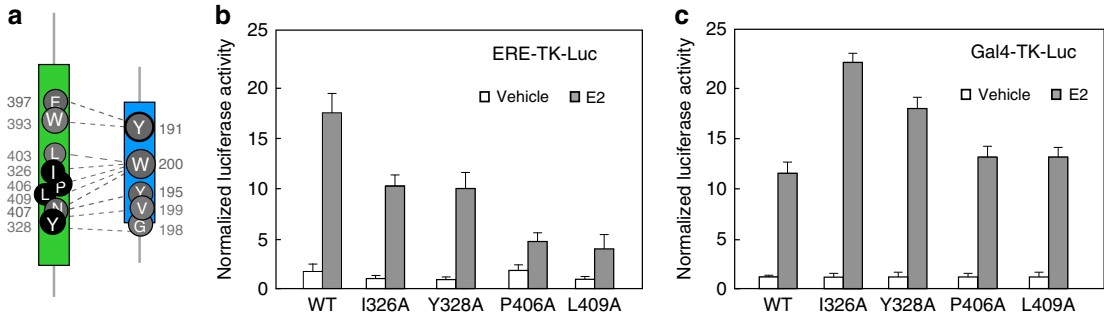

**Fig. 4** Transcriptional regulation of mutations at interfacial LBD residues. **a** Sites of LBD mutation (I326, Y328, P406, and L409), highlighted in black circles. **b–c** Transient transfection reporter activity of the hERα and the Gal4-DBD/hERα-LBD fusion protein, using an ERE-TK-Luc (**b**) or a Gal4-TK-Luc reporter construct (**c**), respectively. Triplicates were carried out and standard deviations were indicated

little change concerning its DNA-binding affinity $K_d = 10.8$ nM, compared to the WT's $K_d = 9.2$ nM (Supplementary Fig. 1d), whereas Y191H increased the binding affinity about threefold to $K_d = 3.2$ nM (Supplementary Fig. 13). Interestingly, this Y191H mutation has also been observed in endometrial cancer samples[41,42]. While its oncogenic function is unknown, we note that the structural role of Y191H with its energetic stabilization of the DBD–LBD interface correlates with increased DNA binding and with elevated transcriptional activity in vitro that may occur in cancer patients.

Finally, we directly tested the potential mutational influence on the observed domain–domain interface itself. The residue N407, part of the hydrophobic cluster among N407, Y195, and W200 at

the interface (Fig. 3d), was mutated into alanine to alter the interfacial structure. The effect of N407A on transcriptional activity was evaluated again by transient transfection reporter assays. In contrast to the Y191H mutation noted above, which increased reporter luciferase activity, the N407A mutation decreases E2-induced reporter activity using both an ERE-TK-Luc and a Gal4-TK-Luc reporter construct (Fig. 3c and Supplementary Fig. 14). Its impact on the interfacial structure was further assessed by tryptophan fluorescence of W200, also part of the domain interface. We emphasize that N407 and W200 are each situated on different domains of the complex. As such, the extent of W200 exposure at the interface—as observed and noted above—is especially critical because the disruption by the

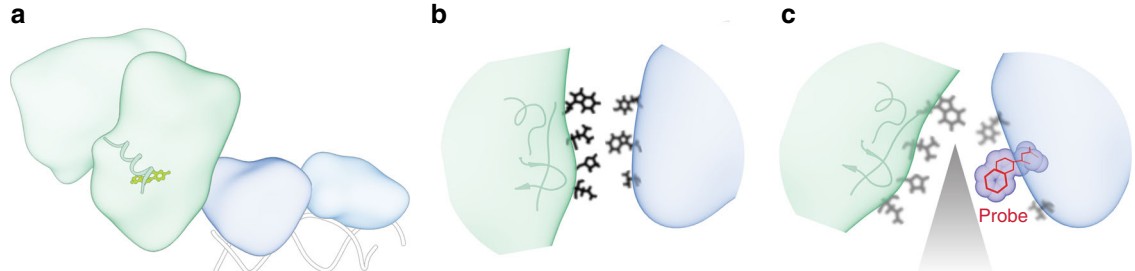

**Fig. 5** Multidomain architecture and cross-talk at the domain interface of the hERα. **a** The hERα homodimeric complex contains both LBDs (green) and DBDs (blue). A hormone ligand (yellow) is capped underneath the LBD's C-terminal helix H12 (ribbon). **b** The LBD–DBD interface consists of the LBD's two β-strands, distant from the ligand-binding pocket and coactivator-binding sites. Disruption of this interfacial cross-talk, which serves as an allosteric channel to transmit the signaling of ligand binding from the LBD to a distant DBD, suppresses hormone-induced transcription. **c** Alteration of the domain cross-talk at the structural level is monitored by intrinsic tryptophan fluorescence (i.e., Trp200 in the middle of the interface as a probe in red blobs), using our genetically engineered hERα construct

N407A mutation would alter the W200 surroundings and reflect the local conformational changes occurring at the domain interface, influencing its emission spectra (Fig. 3e), and thus provide a direct confirmation of domain interaction.

By mutating four other tryptophan residues into phenylalanine and keeping W200 untouched in the hERα$^{complex}$, we were able to utilize this genetically engineered construct for tryptophan fluorescence measurements and to monitor changes to the interfacial structure of W200's surroundings resulting from structural perturbation. A significant decrease was observed in the fluorescence emission spectra for mutant N407A with a peak reduction of about 30% (Fig. 3f). Of note, residue N407 in our structure places the LBD in close contact with W200 of the DBD across the DBD–LBD interface. The reduction in W200 fluorescence in the context of the N407A substitution confirms the observed DBD–LBD interface as necessary for maintaining the appropriate domain interaction and its influence in modulating transcriptional luciferase activity.

## Discussion

The L-shaped boot architecture of hERα complex reported here represents a distinctive spatial arrangement between the DBD and LBD among members of the NR superfamily (Fig. 5a). As schematically illustrated (Supplementary Fig. 15), comparison with existing NR crystallographic or cryo-EM structures shows that hERα$^{complex}$ adopts a different domain organization from currently known NR architectures such as an elongated PPARγ–RXRα[5] and an italicized-X-like RXRα–LXRβ[9] (Supplementary Fig. 15c). The spatial difference is reaffirmed by their structural incompatibility with the experimental scattering data of hERα$^{complex}$, where theoretical SAXS profile of each known NR structure, after threading the hERα sequence (with the ligands kept untouched), yielded a large discrepancy in $\chi^2$ with the experiential scattering data of hERα$^{complex}$ (Supplementary Fig. 16a). At the domain level, a pronounced difference among these NR complexes is indicated by their buried domain surface areas of the LBDs involved in DBD–LBD interactions, with 1219 Å$^2$ for PPARγ–RXRα, 562 Å$^2$ for RXRα–LXRβ, while here we calculated 747 Å$^2$ for hERα (Supplementary Fig. 16b), pointing to a distinguishable contact interface formed between the hERα-DBD and -LBD that is distinct from the others. We note that hERα resembles the extended conformation that USP/EcP adopts[10], although the way individual domains are assembled to interact is different. Specifically, hERα-LBD and -DBD directly interact (Supplementary Fig. 15a), while USP/EcP LBD interacts with DNA instead (Supplementary Fig. 15b). Overall, the

L-shaped boot architecture of the hERα and the allosteric path through the previously unknown DBD–LBD contact presents a distinctive DBD–LBD spatial organization within the NR superfamily.

The functional importance of hERα DBD–LBD contact, based on a favorable interaction between its LBD hydrophobic patch surrounding its two β-strands between helices H5 and H6 (Fig. 5b and Supplementary Fig. 11) is fully supported by the observation that interfacial mutations (e.g., Y191H) alter the receptor's ability to regulate transcription as well as its ability to bind DNA. Of note, a similar β-strand region is also involved in PPARγ–RXRα's domain–domain interactions, where residue F347 from the H5–H6 connecting β-strand region of PPARγ-LBD is shown to functionally mediate its DBD–LBD interface[5], despite an apparent difference from the solution structure[7]. We note that the use of the β-strand region is distinctive for hERα and PPARγ–RXRα, departing from other NR complexes such as RARβ–RXRα and HNF-4α that use a different region near helices H9 and H10 for domain–domain interactions[8,11]. Strikingly, mutations at the LBD hydrophobic patch, residues I326A, Y328A, P406A, and L409A from the β-strand region, significantly inhibited E2-induced transactivation without reducing their capability of hormone and coactivator binding (Fig. 4b–c). Moreover, our structure–function studies find that the hERα DBD–LBD couplings involving the patches in these locations allow for effective signal transmission from the LBD to the DBD. In particular, the DBD–LBD junction allows information about the N407A mutation at the LBD to be allosterically transduced to W200 (Fig. 3f) at the DBD by influencing its surroundings at the interface, as reflected in the reduction of tryptophan fluorescence. As such, alteration of interfacial structures, e.g., by point mutations such as N407A, can be probed via intrinsic tryptophan fluorescence using a genetically engineered hERα construct (depicted in Fig. 5c). In principle, this fluorescence assay can be used to monitor the disruption of the hERα DBD–LBD interface due to binding of novel small molecules, especially because of the increasing importance of drug targeting at protein–protein interfaces[43]. The modulation of the DBD–LBD interface as an "allosteric" channel of hERα with gain or loss of receptor function—going beyond the current focus on the "active" site of estradiol binding—is crucial to the articulation of signaling across the interface and for providing a molecular understanding of the inner-workings of receptor activation.

## Methods

**Recombinant expression and purification.** The human ERα segment containing both the DNA-binding and ligand-binding domain (amino acids 181–552, referred

to as hERα[CDE]) was expressed in *E. coli* cells in the presence of estradiol (E2). The purified hERα protein was incubated with E2, ERE–DNA (5′-TAGGTA-CACGTGACCTGCG-3′ and 5′-CGCAGGTCACTGTGACCTA-3′) and a coactivator TIF2 peptide (KENALLRYLLDKDD) as adopted in the crystal structures available in the literature[12,14,15], referred to as hERα[complex]. The hERα[CDE] was cloned into pMCSG7 vector[44,45]. The primer sequences related to cloning and mutagenesis are shown in Supplementary Table 2. The expression vector with a His-tag was transformed into Rosetta2(DE3)pLysS *E. coli* cells (Novagen). For protein expression, *E. coli* cells were grown in TB medium at 37 °C with 100 μg/ml ampicillin and 34 μg/ml chloramphenicol. When $OD_{600}$ reached 0.4, cell cultures were cooled to 16 °C and protein expression was then induced by the addition of 0.1 mM IPTG in the presence of 10 μM 17β-estradiol (E2). The cultures were shaken at 16 °C for another 18 h before the cells were harvested by centrifugation. The cells were resuspended in a buffer (referred to as buffer A; 50 mM HEPES (pH 7.5), 300 mM NaCl, 50 mM arginine, 50 mM glutamate, 5 mM β-mercaptoethanol (BME), 5% glycerol, 10 μM estradiol, and 10 μM Zn acetate) supplemented with 20 mM imidazole, 0.1 mg/ml DNase I, and protease inhibitor cocktail (Roche, Indianapolis, IN). The cells were disrupted by sonication or using a M110Y microfluidizer (Microfluidics, Newton, MA). Cell debris was removed by centrifugation at 18,000 × *g* for 45 min at 4 °C. Cleared supernatant was incubated with TALON resin (Clontech). Wash by imidazole step gradients and resin elution by buffer A with 40 mM imidazole was applied. Eluted proteins were incubated with TEV protease at a molar ratio of 1:50 (TEV:protein) and dialyzed into buffer A overnight at 4 °C. TEV protease and uncleaved hERα were removed by Talon resin. The protein was subsequently concentrated and purified by HiLoad 16/600 Superdex 200 pg column with an equilibration buffer (referred to as buffer B; 10 mM CHES (pH 9.5), 125 mM NaCl, 5 mM KCl, 4 mM $MgCl_2$, 50 mM arginine, 50 mM glutamate, 5 mM TCEP, 5% glycerol, 10 μm Zn acetate, and 10 μM estradiol E2). The 18-bp oligonucleotide with sequences 5′-TAGGTACACGTGACCTGCG-3′ and 5′-CGCAGGTCACTGTGACCTA-3′ (Integrated DNA Technologies, Inc) contains a consensus estrogen response element (ERE) as adopted in the DBD crystal structure (PDB entry 1HCQ), was heated to 95 °C and slowly cooled down to ensure the formation of a double-stranded DNA duplex (referred to as ERE–DNA). Eluted hERα proteins were incubated with a 1.2× molar ratio of ERE–DNA and a 3.0× molar ratio of the coactivator TIF2 peptide, KHNALL-RYLLDKDD, as adopted in the LBD crystal structure (PDB 1GWR), and placed on ice for 1 h. Final gel filtration purification by a Superdex 200 10/300 GL column (GE) equilibrated with buffer B was performed to obtain the final hERα[complex] samples.

**Hydroxyl radical protein footprinting.** Purified hERα[complex] samples at micromolar concentrations were exposed to a focused synchrotron X-ray white beam for 0–800 ms at the 5.3.1 beamline of Advanced Light Source (Berkeley, CA). The samples were quenched, frozen, and later digested with the protease pepsin. The sites of oxidation were detected and analyzed by liquid chromatography-mass spectrometry. Increasing X-ray exposure time results in an increase in modified population and a reduction in unmodified species (see Supplementary Fig. 2 for representative dose–response plots). The fit to the dose–response plot provides the rate of side chain modification, which is governed by intrinsic reactivity of each amino acid and by the solvent accessibility of the side chain to hydroxyl radicals. The ratio of the intrinsic reactivity and measured footprinting rates yields protection factors (PFs)[22,23], which are used to directly compare the solvent accessibility for different residues across the protein.

Beam parameters were optimized by using an Alexa-488 fluorophore assay. Samples were dialyzed against a footprinting buffer of 5 mM sodium borate, 50 mM NaCl, and 50 mM KCl, pH 9.5, and the protein concentration was adjusted to 2 μM, followed by exposure of 0–800 ms at ambient temperature, and immediately quenched with 10 mM methionine amide to prevent secondary oxidation. Protein samples were then treated with 10 mM DTT at 56 °C for 45 min and alkylated with 25 mM iodoacetamide at room temperature in the dark for 45 min, and then formic acid was added to a final concentration of 0.5% to adjust the target pH = 2. Proteolytic cleavage of the irradiated samples was performed using pepsin (Promega, Inc.) at 37 °C for 3 h at an enzyme-to-protein molar ratio of 1:20. The digestion reaction was terminated by heating at 95 °C for 2 min. Identification and quantification of the sites of radiolytic modification were performed by liquid chromatography-mass spectrometry (LC-MS) analysis of pepsin-digested samples on an Orbitrap Elite mass spectrometer (Thermo Scientific, CA) interfaced with a Waters nanoAcquity UPLC system (Waters, MA). A total of 2 pmol of proteolytic peptides were loaded on a trap column (180 μm × 20 mm packed with C18 Symmetry, 5 μm, 100 Å (Waters, MA)) to wash away salts and concentrate peptides. The peptide mixture was eluted on a reverse phase column (75 μm x 250 mm column, packed with C18 BEH130, 1.7 μm, 130 Å (Waters, MA)) using a gradient of 2–55% mobile phase B (0.1% formic acid and acetonitrile (ACN)) vs. mobile phase A (100% water/0.1% formic acid) over a period of 60 min at 37 °C with a flow rate of 300 nl/min. The peptides eluted from the reverse phase column were introduced into the nano-electrospray source at a capillary voltage of 2.5 kV. Tandem mass spectrometry (MS/MS) data were acquired in the positive ion mode. In the first MS (MS1) analysis, a full scan was recorded for eluted peptides (*m/z* range of 350–1600) in the Fourier transform mass analyzer at resolution of 120,000, followed by MS/MS of the 20 most intense peptide ions scanned in the ion trap

mass analyzer. Detected ion currents for peptic peptides in MS1 experiments were used to determine the extent of oxidation for each modified site by separate quantification of the unmodified peptides and their radiolytic products, and MS/MS spectra were acquired to identify specific sites of modification. The resulting MS/MS spectra against the hERα[CDE] protein database using the software MassMatrix[46] with mass accuracy values of 10 ppm and 0.7 Daltons for MS1 and MS/MS scans, respectively, and allowed variable modifications including carbamidomethylation for cysteines and all known oxidative modifications previously documented for amino acid side chains. All MS/MS spectra for reported sites were examined manually and verified individually. The footprinting rate ($k_{fp}$) was derived for each residue lying on individual domain surfaces via a dose–response curve, i.e., the fraction of unmodified residues by hydroxyl radicals as function of X-ray exposure time. Single-residue protection factors (PFs) were subsequently calculated by dividing the intrinsic reactivity of the residue by its $k_{fp}$ value[22,23].

**Small-angle X-ray scattering.** A chromatography-coupled setup was used for SAXS data collection of purified hERα[CDE] proteins eluted with the ligands of E2, ERE–DNA, and a coactivator TIF2 peptide at the BioCAT-18-ID beamline of the Advanced Photon Source (Argonne, IL). The hERα[complex] was eluted through a size exclusion column (SEC) equilibrated with saturated ligands. A Superdex 200 10/300 column (GE) at a flow rate of 0.5 ml/min was used in conjunction with an AKTA pure FPLC machine (GE Health Sciences). Scattering images were collected every 3 s along the elution at a flow rate of 0.5 ml/min. Each image was recorded with a 2-s exposure time. A set of six scattering images of the hERα[complex] near the elution peak were merged and a total of 34 images before and after the peak were used as buffer scattering for buffer subtraction. Data reduction resulted in a final one-dimensional $I(q)$ profile with a bin size of $\Delta q \sim 0.004$ Å$^{-1}$. A buffer of 10 mM CHES pH 9.5, 50 mM NaCl, 50 mM KCl, 4 mM $MgSO_4$, 50 mM Arg, 50 mM Glu, 5% glycerol, 1 mM TCEP, and 2 μM E2 was used for size exclusion chromatography. The X-ray energy was 12 keV. Parameters for SEC-SAXS data collection are listed in Supplementary Table 3, according to the recent SAXS data deposition guideline and practice[47,48].

**Computational docking.** Docking simulations were based on the crystal structures of the DNA-bound DBD homodimer and the E2-bound LBD homodimer in complex with a coactivator TIF2 peptide by a series of rigid-body docking and coarse-grained modeling, with extensive conformational search[19,29], followed by atomic-level molecular dynamics simulations restrained between the DBD packet of Y191/Y195/W200 and the LBD packet of I326/W393/L409. Each docked conformation was evaluated against experimental SAXS and footprinting data via the scoring functions $\chi^2$ (Eq. 1) and $\varphi^2$ (Eq. 2) for the selection of the best-fit ensemble structures of the hERα[complex].

Crystal structures of the DNA-bound DBD homodimer (PDB entry 1HCQ) and the E2/peptide-bound LBD homodimer (PDB entries 1QKU and 1GWR) were used in the following three steps. Rigid-body docking of these two domain-homodimers each treated as a separate entity was performed to generate an initial set of 3125 poses that uniformly cover the interdimer rotational degrees of freedom. Coarse-grained (CG) Langevin simulations were implemented for each pose to extensively sample the translational motion between the centers of mass of the two dimers with a distance range of 25–50 Å, for a total of 390,625 ns. Final atomic-level structure reconstruction was performed by aligning the crystal structures of the two domain-dimers onto the CG structures for those with a center-of-mass distance of <20 Å between the clusters of footprinting-detected residues Y191/Y195/W200 and I326/W393/L409. The domain-connecting hinge was built using the loop modeling software Jackal[49]. Restrained simulations were performed using Amber16[50], whereas the LBD homodimer was position restrained and the DBD homodimer was RMSD restrained both at their Cα and Cβ atoms with a harmonic spring constant of 10.0 kcal/mol/Å$^2$. To arrive at a set of 176 final conformations with the lowest $\chi^2$ (Eq. 1) and $\varphi^2$ (Eq. 2), the center-of-mass distance of the Cγ atoms was linearly restrained between the two clusters of residues (Y191/Y195/W200 and I326/W393/L409) with a target distance moving from 10 to 2 Å over a period of 1.0 ns and a force constant of 20 kcal/mol/Å$^2$.

The molecular force fields of amber-ff14SB[51], TIP3P[52], and DNA.OL15[53] were used for all-atom, explicit-solvent simulations, and the parameters for the estradiol E2 were generated using the software package antechamber[54]. The system was placed in a rectangle water box with a buffer distance of 10 Å in the presence of a 150-mM salt solution. Standard periodic boundary conditions were applied with a non-bond cutoff of 12 Å. Simulations were performed at a temperature of 300 K and a pressure of 1 atm with a 2-fs time step.

**Transient transfection reporter assay.** Wild-type and mutant hERα-LBD (residues 303–595) were subcloned with Gal4-DBD to generate a Gal4-DBD/hERα-LBD fusion construct. The sequences of primers for cloning and mutagenic sequences are shown in Supplementary Table 2. The effects of ERα mutations on transcription activity were evaluated by transient transfection reporter assays using a dual luciferase reporter assay. HeLa cells were co-transfected with an HA-hERα or a Gal4-DBD/hERα-LBD expression plasmid (2 μg) on 60-mm plate and an ERE-TK-Luc or Gal4-TK-Luc (2.5 μg) and Renilla-Luc (0.5 μg). Cells were then split to a 24-well plate and starved at 2% FBS in DMEM medium. After overnight starvation,

medium was replaced with or without 100 nM E2 (Cayman #10006315). Twelve hours later, cells were harvested and firefly luciferase (FLuc) and renilla-luciferase (RLuc) activities were measured using dual luciferase reporter assay kit (Promega, E1910) according to the manufacturer's protocol. An aliquot of lysates was subject to western blotting to visualize the expression of wild-type and mutant hERα, where western blottings were performed using anti-HA (1:1000, sc-805, Santa Cruz), anti-Gal4 (1:2000, sc-729, Santa Cruz), and anti-β-actin (1:1000, A5441, Sigma-Aldrich) antibodies. Briefly, 50 µl of lysate was mixed with 50 µl of Luciferase Assay Reagent II to determine luminescent signal for FLuc. After the luminescence was quantified, the FLuc activity was quenched and RLuc activity was measured by adding 5 µl Stop & Glo Reagent (E1910 Promega). Luciferase activity was normalized to the level of RLuc activity. Each reaction was performed in triplicate, and triplicates were averaged prior to statistical analysis.

**Genetically engineered hERα-specific fluorescence assay.** Four out of all five tryptophan residues (W292, W360, W383, and W393) in the hERα$^{complex}$ were mutated to phenylalanine except W200, which was kept as an intrinsic fluorescence probe. The emission spectra of hERα$^{complex}$/W200 were recorded between 310 and 400 nm with a bandwidth of 5 nm at 25 °C using a FluoroMax-3 spectrofluorometer (Horiba Scientific). Buffer correction was applied to all samples at a protein concentration of 0.1 mg/ml. Excitation at 295 nm was used to minimize the influence from tyrosine.

**Fluorescence anisotropy DNA-binding assay.** The fluorescence-conjugated double-strand ERE–DNA was prepared by annealing 6-FAM (6-carboxy-fluorescein) 5′-labeled strands (5′-T**AGGTCA**CAG**TGACCT**GCG-3′ and 5′-CGC**AGGTCA**CTG**TGACCT**A-3′; IDT, Inc) in a buffer containing 10 mM CHES (pH 9.5), 50 mM KCl, 50 mM NaCl, 4 mM MgCl$_2$, 50 mM arginine, 50 mM glutamic acid, 5 mM TCEP, 5% glycerol with 10 µM ZnCl$_2$, and 10 µM E2. The resulting 20 nM ERE–DNA was incubated with purified proteins in the presence of E2 and coactivator peptides for 10 min for the binding assay and loaded into a 96-well plate (Greiner Bio-one). Fluorescence anisotropy intensity was recorded at a series of hERα$^{CDE}$ protein concentrations using a Tecan M1000-PRO microplate reader.

**Surface plasmon resonance.** Peptide binding between the hERα$^{CDE}$ and a coactivator TIF2 peptide (KENALLRYLLDKDD) was measured by surface plasmon resonance (SPR) using a Biacore T100 system (GE Healthcare). Sensorgrams were recorded for a concentration series of the hERα$^{CDE}$–DNA-E2 complex, where the biotinylated TIF2 peptide (captured at 10 RUs) was immobilized on an SA sensor chip (GE Healthcare) and a flow rate of 20 µl/min of the complex at a concentration range of 0–5 µM was used for injection over the peptide-binding surface. Measurements were conducted at 25 °C using a Biacore T100 system.

## Data availability

SEC-SAXS data and ensemble structures determined by our multi-technique iSPOT structural-biology platform have been deposited in the Small Angle Scattering Biological Databank[55] (SASDB access code SASDDU8; https://www.sasbdb.org/data/SASDDU8). Other data are available from the corresponding author on reasonable request.

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

## Acknowledgements

We thank Krishna M. Ravikumar for contributing to early stage computations, Witold Surewicz for helping fluorescence measurements, Geof Greene for inspiration of this project, and Marc Parisiaen for discussion. Supported by NIH grants GM114056 (S.Y.), EB009998 (M.R.C.), and UL1TR000439 (P.B. Davis). Use of the synchrotron sources was supported by the U.S. Department of Energy (DE-AC02-06CH11357 and DE-AC02-98CH10886) and by the NIH (GM103622).

## Author contributions

S.Y. designed the research. W.H. and Y.P. expressed and purified the complex. W.H., S.C. and S.Y. collected scattering data. S.Y. performed integrative multi-technique modeling. J.K., S.G., C.R., W.H. and S.Y. collected and analyzed footprinting data. W.H. performed fluorescence anisotropy. W.H. and Y.C. performed SPR. X.Z., A.A., D.M. and H.-Y.K. performed reporter assays. Y.P. performed tryptophan fluorescence. S.Y., M.R.C. and H.-Y.K. wrote the manuscript with the inputs from all authors.
