## [Peer Review File · Nature Communications]

Reviewers' comments:

Reviewer #1 (Remarks to the Author):

The paper reports a study of the structure-function relationships in hER α , an important nuclear hormone receptor. The work uses an integrative approach that combines small-angle X-ray scattering (SAXS), hydroxyl radical footprinting and computational modeling. It brings a new solution structure to the gallery.

The authors claim the discovery of a novel asymmetric L-shaped "boot" structure and the finding of the specific sites involved in DBD-LBD interactions, the results being validated by few site-specific mutagenesis experiments.

The overall approach and the correlations methods are appropriate. The results are interesting, pointing to a novel contact between LBD and DBD. A weakness of the paper is the over interpretation of experimental data at some places.

Some conclusions like the novelty of the architecture are surprising. How the paper's contribution can help drug design is also not clear.

Preliminary remark: it is difficult for the reader to identify within the complex the position or the relative position of the structural elements, notably the amino-acids implicated in the contacts. One would like to see a figure of the complex where the helices are shown with the known DBD/LBD nomenclature of the critical ones on the drawing.

Questions and comments with focus on the structural aspects:

1. DNA: what is the size and the sequence of the samples ? A view down the DNA helix would allow to see the relative positions of DBDs and LBDs. It would help to clarify the comparisons with other structures.

2. SAXS fig 2d: the fit of the curve calculated from the best-fit structure to the experimental data is not convincing. The gyration radius of the species in solution seems to be larger than the one calculated from the model. Thus a Guinier plot should be depicted in an insert. Furthermore, the fitness of the calculated curve to the experimental data should be depicted as residuals as a function of Q. This is a complementary check together with the χ^2 value.

What is the effect of the large C α -atom RMS fluctuation (C α -RMSF) observed at the domain-connecting region (D region) on the χ^2 value. Fig 2 (b and c) shows one connection only.

3. Comparison with other structures:

-The resolution of the model does not allow a meaningful comparison of the interfaces surfaces with the ones observed in the high resolution crystal structures (moreover the dynamics of the two systems is different). The differences are anyway not large.

-The L shape asymmetric architecture is not novel. The most relevant comparisons would be with HNF4, another homodimer, and USP-EcR for the DNA target aspect (3bp inverted repeat). As for the so called "boot-shape" the cryoEM structure of USP-EcR is a good candidate for a comparison. A look at both the SAXS and the EM envelopes suggest a comment. A comparison of the structures at the C α level will highlight the differences and similarities.

Reviewer #2 (Remarks to the Author):

The manuscript entitled "Multidomain architecture of estrogen receptor reveals interfacial cross-talk between its DNA-binding and ligand-binding domains" describes the use of multiple biophysical techniques to identify the interactions between the DBD and LBD domains of hER α . The

authors use hydroxyl radical footprinting, SAXS, and computational modeling to report on a structure of multidomain hER α . The structure was further validated using site directed mutagenesis along with activity assays and Trp fluorescence. The approach used by the authors was very elegant utilizing the strengths of the individual techniques in conjunction for prediction of the multidomain structure. In addition, from the resultant data, a novel allosteric pathway in the nuclear receptor superfamily was determined. The experiments were designed well and the proper controls were run. The statistical analysis for computational modeling was well done. The manuscript should be published with minor revisions in the language as described below.

On page 9 "While their oncongenic relevance remains to be established, alternation" The word should be alteration not alternation

On the top of page 10 Fig.. S11 is incorrectly cited as Fig. S1

Reviewer #3 (Remarks to the Author):

The manuscript by Wei Huang et al. reports a result of structural estimation of DBD-LBD complex domain of hER α . Two or more experiments and calculation technique is applied to analysis, and they are utilized fully-complementarily. Molecular arrangement of this complex is presumed in SAXS- computational-docking approach based on the high contrast results with several clear amino acids shown in identification of the binding interface by MS. Furthermore, the result of the mutation analysis in an interacting region also have good consistent on the consideration of a molecular arrangement. However, I consider that SAXS data and description are especially insufficient.

1. You have to follow the newest publication guideline on SAXS data.

Jill Trehwella et al. Acta. Crystallogr. D (2017) 73, 710-728.

"2017 publication guidelines for structural modelling of small-angle scattering data from biomolecules in solution: an update"

Measurement condition and several physical quantity derived from the result of analysis, for example, R_g , $I(0)$, MW, etc. should be shown at least. Those values will act as important factor in order to demonstrate the justification of experimental data and analysis.

2. I propose your submitting the resultant structure to SASBDB.

<https://www.sasbdb.org/>

The direct check on each PC's desktop will help to good understanding for the readers. Since you will be able to get SASBDB code like PDB code, you will show it to the text.

3. The fit for the doking-model in figure 2d using data at higher angle region that is quite noisy and it is surprising the chi-squared values are satisfactory. In fact, there is a wiggle in the region over $q=0.15$ that would be difficult to fit between the experimental and theoretical profiles. Of course, we know that it is difficult to acquire the high S/N data at the wider-angle region in SEC-SAXS because of the concentration dilution. In such a case, the resultant value, for example, R_g , $I(0)$, MW, D_{max} , etc... will support the total understanding of the result for the readers. The form factor of this complex had better be measured up to $q \times R_g \sim 8$. Because R_g value is not shown, I could not confirm that the measurement angle region is enough for the speculation of the molecular shape. Moreover, it is quite difficult to estimate the structure of amino acids although the molecular surface shape measured at $q=0.1 \sim 0.25$ will restrict the structure of amino acids which form the molecular surface. You had better reference the following article of VDR (DOI: 10.1021/acs.jmedchem.6b00682).

4. I consider that the discussion about the high/low of the absolute value of LogPF is quite difficult except for the values of several residues like W200. Therefore, I propose that it had better be

discussed by not “high/low” but “positive/negative” if I did not misunderstand about it.

5. What is “MS/MS” in line 9, Page 4?

6. Which is the correct, “Y197” in figure 1d or “H197” in the text of page 4?

Reviewer #4 (Remarks to the Author):

This manuscript proposes inter-domain interaction between DBD and LBD of hER α , which is also observed in other members of nuclear receptor family. Here, they combined three different techniques to identify the sites of inter-domain interaction and propose an asymmetric L-shaped boot structure, which is novel for hER α . To further validate the model, the authors mutate the sites of interaction and showed the change in ER transcriptional activity by luciferase reporter assays. Overall, the manuscript is clearly written, and the results support their model. However, their proposed model would be further strengthened by careful characterization of the DBD-LBD interaction site mutants as mentioned below.

1. Several mutations were generated at the proposed interaction sites in LBD as well as DBD which showed diminished transcriptional activity (FigS10), which they think might be due to loss of inter-domain interaction. However, the authors have not shown any biochemical data on how these mutations could affect ER's ability to bind ligand as that could also lead to reduced transcription activity. Similarly, these mutations could not only affect hormone binding but could affect allostery required to connect ligand binding with coregulator interaction. This could be tested by fusing the ER LBD to a Gal4DBD and looking for impact to transactivation.

2. In validation of the L-shaped model, the authors showed a good fit of experimental SAXS data with the computational docked model (figure 2d). However, the SAXS scattering data for DBD-LBD interaction interface mutants is not provided. This data would further strengthen their model if they could show that one or two mutant protein SAXS profiles do not fit well with WT (and reflect conformational change), especially in case of N407A mutant.

3. In the current manuscript authors proposed a novel L-shaped architecture of hER α , which they think differs from other members of nuclear receptor family (and other ER α structures estimated via SAXs). However, more discussion should be included comparing their model and its functional relevance with the DBD-LBD inter-domain architecture seen in other nuclear receptors. I would also suggest a good blob model of domain arrangement as final figure would enhance the clarity and understanding of the system to the readers.

Reply to Referees

NCOMMS-18-05656-T

“Multidomain architecture of estrogen receptor reveals interfacial cross-talk between its DNA-binding and ligand-binding domains”

We thank the referees for their constructive comments. We have addressed all the questions raised and the suggested changes have substantially improved the manuscript. In the following reply, the reviewers' comments are shown in black and our responses follow and are shown in blue. The corresponding edits and changes are highlighted in blue in the manuscript.

Reviewer #1 (Remarks to the Author):

The paper reports a study of the structure-function relationships in hER α , an important nuclear hormone receptor. The work uses an integrative approach that combines small-angle X-ray scattering (SAXS), hydroxyl radical footprinting and computational modeling. It brings a new solution structure to the gallery.

The authors claim the discovery of a novel asymmetric L-shaped “boot” structure and the finding of the specific sites involved in DBD-LBD interactions, the results being validated by few site-specific mutagenesis experiments.

The overall approach and the correlations methods are appropriate. The results are interesting, pointing to a novel contact between LBD and DBD. A weakness of the paper is the over interpretation of experimental data at some places.

Some conclusions like the novelty of the architecture are surprising.

In our revised manuscript, we have now included a new figure (Suppl. Figure 15) to provide a comparison of the architectures available for several nuclear receptor complexes containing both LBD and DBD. We have edited the text concerning their similarities and differences accordingly (p. 12 and p13).

How the paper's contribution can help drug design is also not clear.

We have added a schematic representation (Figure 5b and Figure 5c) to illustrate the interfacial targeting by using the identified LBD-DBD interface to help drug design. The newly identified interface is of importance since it has been recognized that targeting the interfaces between proteins has significant therapeutic potential (see Nature 450(7172):1001-9, 2007; “Reaching for high-hanging fruit in drug discovery at protein-protein interfaces”; a new reference added - ref. 43). In the context of hER α , the DBD-LBD interface (depicted in Fig. 5b) can be targeted by small molecules, in which the interfacial mutation, already shown to alter the interface, can serve as a control or proof-of-principle study (Fig. 3f). In particular, disruption of the interface can be conveniently probed by tryptophan fluorescence (as depicted in Fig. 5c). While drug discovery itself is beyond the scope of this manuscript, we have added some discussion in the main text (p. 4 and p. 14).

Preliminary remark: it is difficult for the reader to identify within the complex the position or the relative position of the structural elements, notably the amino-acids implicated in the contacts. One would like to see a figure of the complex where the helices are shown with the known DBD/LBD nomenclature of the critical ones on the drawing.

We have followed the established DBB/LBD nomenclature of ref. 15 in describing the architecture. A cartoon/12-helix figure (Suppl. Figure 11; also shown on the right) is now included to show the relative positioning of these amino acids that are involved in the domain-domain contacts. We have also clarified this point accordingly in the main text (p. 10).

Questions and comments with focus on the structural aspects:

1. DNA: what is the size and the sequence of the samples?

We have included the 18-bp DNA-ERE (5'-TAGGTACACGTGACCTGCG-3' and 5'-CGCAGGTCACTGTGACCTA-3') in the main text and in the Methods (p. 5 and p. 15).

A view down the DNA helix would allow to see the relative positions of DBDs and LBDs.

A figure with a view down the DNA is shown in Fig. 2c with a 90-degree rotation of the domain arrangement shown in Fig. 2b.

It would help to clarify the comparisons with other structures.

This is a very constructive suggestion (as reviewer #4 also pointed out). A new figure (Suppl. Figure 15) is added to illustrate the overall architectures of currently known members of the nuclear receptor family (including USP/EcP as mentioned below). Accordingly, we have included extensive discussion about their comparison (p. 13).

It is striking that hER α and USP/EcP are similar in both adopting an extended L-shape (Suppl. Figure 15a-b), although there is a major difference with regard to the residues involved at the LBD-DBD interface. In the USP/EcP, the helix H9 of USP-LBD is involved to form the functional interface, but with DNA, as opposed to the DBD as observed in the hER α . The RAR β -RXR α heterodimer uses the loop region between H9 and H10 helices of RAR β -LBD to interact with its own DBD. Notably, despite the helix H9 being involved in the domain-domain contacts of both RXR α -LXR β and HNF-4 α , the RXR α -LXR β domain interface is not as tight as HNF-4 α (via their contact surface areas), as demonstrated in Suppl. Figure 16. In contrast, hER α mostly use the hydrophobic patch from the two beta-strands between helices H5 and H6 (see the nomenclature in Suppl. Figure 11). Strikingly, a similar beta-strand region is involved in the LBD-DBD interface of PPAR γ -RXR α , where a functionally critical residue Phe347 at the interface is from the H5-H6 connecting beta-strand region of PPAR γ LBD. It should be noted that despite this similarity, the PPAR γ -RXR α domains assemble into a fairly compact overall architecture, compared to the extended L-shape that hER α and USP/EcP adopt. Finally, we observed that both hER α and USP/EcP bind inverted repeat (IR) DNA, although other nuclear receptors mentioned here bind direct repeat (DR) DNA. We have included additional discussion to reflect these points (p. 13 and p. 14).

2. SAXS fig 2d: the fit of the curve calculated from the best-fit structure to the experimental data is not convincing. The gyration radius of the species in solution seems

to be larger than the one calculated from the model. Thus a Guinier plot should be depicted in an insert.

We have inserted a Guinier plot into the upper corner in a new Fig. 2d. We also updated the figure caption accordingly (inserted is a Guinier plot of $\log(q)$ as a function of q^2), which gives the radius of gyration $R_g = 38.0 \pm 0.3 \text{ \AA}$.

Furthermore, the fitness of the calculated curve to the experimental data should be depicted as residuals as a function of Q^2 . This is a complementary check together with the χ^2 value.

We calculated the residuals after subtraction and plotted them as black dots at the bottom of Figure 2d

What is the effect of the large $C\alpha$ -atom RMS fluctuation ($C\alpha$ -RMSF) observed at the domain-connecting region (D region) on the χ^2 value.

We have included a new Suppl. Figure S8 (also shown below) for both $C\alpha$ -RMSF and χ^2 as a function of the loop flexibility via RMSD. The overall $C\alpha$ -RMSF values are in the range of 10-30 \AA (Suppl. Figure 8a), which is consistent with the nature of the high flexibility of these loop regions as expected. In particular, the loop region can have fairly large flexibility (with a RMSD value up to 23 \AA) (Suppl. Figure 8b), although its contribution to overall scattering is relatively small with the overall $\chi^2 = 1.2 \pm 0.1$ (Suppl. Figure 8b). These results show the structural fluctuation in the loop contributes $<10\%$ to the overall χ^2 value of the entire complex. This suggests that the overall fitting is more sensitive to domain-arrangements (or orientations) between LDBs and DBDs, as opposed to the hER α 's loop conformations despite the loop flexibility observed. This is in part due to (a) the ensemble-averaging of loop structures and (b) the dominating SAXS signals arising from the distance separation between two large groups (i.e., LDBs and DBDs). We have included additional discussion to reflect these points (p. 9).

Fig 2 (b and c) shows one connection only.

In new Figure 2, the ensemble-structures of connecting loops are now shown.

3. Comparison with other structures:

-The resolution of the model does not allow a meaningful comparison of the interfaces surfaces with the ones observed in the high resolution crystal structures (moreover the dynamics of the two systems is different). The differences are anyway not large.

We agree with the reviewer that comparison of our hER α ensemble-structures with other nuclear receptor (NR) complexes is valuable. As shown in Suppl. Figure 15, it is quite clear that the interface is near the beta-strand region between helices H5 and H6 in the hER α (see the schematic drawing about the positioning using a canonical 12-helix notation in Suppl. Figure 11).

This is different from other complexes that involve helices H9/H10 as in RXR α -LXR β and HNF-4 α , as described above.

In addition, we have used molecular dynamics (MD) simulations to monitor the interfacial dynamics of our post-modeling structural analysis. These all-atom simulations allow us to examine the deformation of surface residues upon the complex formation of the hER α . This is illustrated by overlapping the crystal structures of individual domains with our ensemble-structures (see a new Suppl. Figure 9; also on the right). The results demonstrate modest conformational changes in the interfaces compared to the crystal structures of individual domains, especially for the DBD surface. We have included discussion to reflect this point (p. 9).

-The L shape asymmetric architecture is not novel. The most relevant comparisons would be with HNF4, another homodimer, and USP-EcR for the DNA target aspect (3bp inverted repeat). As for the so called “boot-shape” the cryoEM structure of USP-EcR is a good candidate for a comparison. A look at both the SAXS and the EM envelopes suggest a comment. A comparison of the structures at the C α level will highlight the differences and similarities.

We agree with the reviewer that there is a similarity to the cryoEM model of USP-EcP, as noted above. We have included a new citation (ref. 10; p. 2) and supplied a new figure (Suppl. Figure 15a-b; also shown below). Specifically, the similarity is the overall extended conformations (e.g., L-like shape) and the major difference is on the LBD-DBD interaction. In the hER α , LBD directly interacts with DBD, while there is no LBD-DBD interaction in USP-EcP (where LBD interacts with DNA instead). Additional discussion is included in the text (p. 4 and p. 13) and in the abstract.

Reviewer #2 (Remarks to the Author):

The manuscript entitled “Multidomain architecture of estrogen receptor reveals interfacial cross-talk between its DNA-binding and ligand-binding domains” describes the use of multiple biophysical techniques to identify the interactions between the DBD and LBD domains of hER α . The authors use hydroxyl radical footprinting, SAXS, and computational modeling to report on a structure of multidomain hER α . The structure was further validated using site directed mutagenesis along with activity assays and Trp fluorescence. The approach used by the authors was very elegant utilizing the strengths of the individual techniques in conjunction for prediction of the multidomain structure. In addition, from the resultant data, a novel allosteric pathway in the nuclear receptor superfamily was determined. The experiments were designed well and the proper controls were run. The statistical analysis for computational modeling was well done. The manuscript should be published with minor revisions in the language as described below.

On page 9 “While their oncogenic relevance remains to be established, alternation”
The word should be alteration not alternation

We have made the correction

On the top of page 10, Fig. S11 is incorrectly cited as Fig. S1

We have fixed the typo.

Reviewer #3 (Remarks to the Author):

The manuscript by Wei Huang et al. reports a result of structural estimation of DBD-LBD complex domain of hER α . Two or more experiments and calculation technique is applied to analysis, and they are utilized fully-complementarily. Molecular arrangement of this complex is presumed in SAXS- computational-docking approach based on the high contrast results with several clear amino acids shown in identification of the binding interface by MS. Furthermore, the result of the mutation analysis in an interacting region also have good consistent on the consideration of a molecular arrangement. However, I consider that SAXS data and description are especially insufficient.

1. You have to follow the newest publication guideline on SAXS data.

Jill Trehwella et al. *Acta. Crystallogr. D* (2017) 73, 710-728.

“2017 publication guidelines for structural modelling of small-angle scattering data from biomolecules in solution: an update”

Measurement condition and several physical quantity derived from the result of analysis, for example, R_g , $I(0)$, MW, etc. should be shown at least. Those values will act as important factor in order to demonstrate the justification of experimental data and analysis.

Following the suggestion, we have included a new table (Suppl. Table 2) that summarizes the beamline parameters used for our SEC-SAXS data collection at the APS BioCAT-18ID, including wavelength, photon flux, R_g , and $I(0)$, MW, q-range, exposure time, and temperature. We have referred to this table in the Methods (p. 15). In addition, as described below, all the parameters have been deposited and approved by the SASBDB.

2. I propose your submitting the resultant structure to SASBDB. <https://www.sasbdb.org/>

The direct check on each PC's desktop will help to good understanding for the readers. Since you will be able to get SASBDB code like PDB code, you will show it to the text.

We have submitted our SAXS data/parameters used for SEC-SAXS and our ensemble-structures into the SASBDB (access code **SASDDU8**; a screenshot on the right).

This deposit has been approved by the SASBDB. It is not in the public domain yet, but available for journal referees and editors. This can be accessed via the URL (<https://www.sasbdb.org/data/SASDDU8/wai7yj7q2t>).

The SASBDB code is now described in Methods (p. 16) and referred to in the figure captions (Figure 2b, Suppl. Figure 3, and Suppl. Figure 15a). Again, this has been a valuable suggestion.

3. The fit for the docking-model in Figure 2d using data at higher angle region that is quite noisy and it is surprising the chi-squared values are satisfactory. In fact, there is a wiggle in the region over $q=0.15$ that would be difficult to fit between the experimental and theoretical profiles. Of course, we know that it is difficult to acquire the high S/N data

at the wider-angle region in SEC-SAXS because of the concentration dilution. In such a case, the resultant value, for example, R_g , $I(0)$, MW, D_{max} , etc... will support the total understanding of the result for the readers. The form factor of this complex had better be measured up to $q \times R_g \sim 8$. Because R_g value is not shown, I could not confirm that the measurement angle region is enough for the speculation of the molecular shape. Moreover, it is quite difficult to estimate the structure of amino acids although the molecular surface shape measured at $q=0.1\sim 0.25$ will restrict the structure of amino acids which form the molecular surface. You had better reference the following article of VDR (DOI: 10.1021/acs.jmedchem.6b00682).

Given the approval of our SEC-SAXS data deposit in the SASBDB, the fitting of SAXS data yields the chi-square values in the range of 1.1 to 1.3. The fitting results can be accessed at the URL (<https://www.sasbdb.org/data/SASDDU8/wai7yj7q2t>), and also shown below (on the right is a representative structure from our ensemble-structures).

Based on the SASBDB report, the R_g value is $38.0 \pm 0.3 \text{ \AA}$. We also included this R_g value in the figure caption of Figure 2. Given the upper bound of the q -range of 0.25 \AA^{-1} , the resulting $q \times R_g = 9.5$. We have cited the reference mentioned and other SASBDB related references (ref. 44, ref. 45, and ref. 47), which has been used to guide the reporting of Suppl. Table 2.

Now given the approval of our SASBDB deposit, the chi-square values (in the range of 1.1 to 1.3), obtained from CRY SOL, have been deposited. These data now can be accessed at the URL

(<https://www.sasbdb.org/data/SASDDU8/wai7yj7q2t>), which is available for journal referees and editors but not in the public domain yet). A screen shot is shown on the right to illustrate the overall fitting of SAXS data and a representative structure.

JSmol

4. I consider that the discussion about the high/low of the absolute value of LogPF is quite difficult except for the values of several residues like W200. Therefore, I propose that it had better be discussed by not “high/low” but “positive/negative” if I did not misunderstand about it.

The high/low value of logPF is rather relative. Its correlation with the solvent accessible surface area, as indicated by the dashed line in Figure 1f, would help understand the role for structural interpretation. We have further made clarification in the main text (p. 5).

5. What is “MS/MS” in line 9, Page 4?

We changed “MS/MS” to “tandem MS (MS/MS)”.

6. Which is the correct, “Y197” in figure 1d or “H197” in the text of page 4?

We have fixed it now in the main text (H197 should read as Y197).

Reviewer #4 (Remarks to the Author):

This manuscript proposes inter-domain interaction between DBD and LBD of hER α , which is also observed in other members of nuclear receptor family. Here, they combined three different techniques to identify the sites of inter-domain interaction and propose an asymmetric L-shaped boot structure, which is novel for hER α . To further validate the model, the authors mutate the sites of interaction and showed the change in ER transcriptional activity by luciferase reporter assays. Overall, the manuscript is clearly written, and the results support their model. However, their proposed model would be further strengthened by careful characterization of the DBD-LBD interaction site mutants as mentioned below.

1. Several mutations were generated at the proposed interaction sites in LBD as well as DBD which showed diminished transcriptional activity (FigS10), which they think might be due to loss of inter-domain interaction. However, the authors have not shown any biochemical data on how these mutations could affect ER's ability to bind ligand as that could also lead to reduced transcription activity. Similarly, these mutations could not only affect hormone binding but could affect allostery required to connect ligand binding with coregulator interaction. This could be tested by fusing the ER LBD to a Gal4DBD and looking for impact to transactivation.

We greatly appreciate the reviewer's suggestions. Accordingly, we have generated wild-type and mutant Gal4-DBD/hER α -LBD expression constructs followed by transient transfection reporter assays. In the previous version, our data showed that ER mutants, I326A, Y328A, P406A and L409A, significantly lost E2-induced reporter activity (old Suppl. Figure 10). Our new data show that Gal4-DBD/hER α -LBD mutants still retain their ability to mediate E2-induced reporter activity (new Fig. 4c; also see below). These results indicate that these LBD mutants possess comparable hormone binding and coactivator recruitment activity to that of the wild-type hER α . Overall, our data support the allosteric role of the DBD-LBD interface for E2-induced transcriptional regulation. We have included these results (p. 10) and added related discussion (p. 14), as well as in Methods (p. 16 and p21). For clarity, we moved old Fig. S10 to new Fig. 4b (see below).

2. In validation of the L-shaped model, the authors showed a good fit of experimental SAXS data with the computational docked model (figure 2d). However, the SAXS scattering data for DBD-LBD interaction interface mutants is not provided. This data would further strengthen their model if they could show that one or two mutant protein SAXS profiles do not fit well with WT (and reflect conformational change), especially in case of N407A mutant.

We agree with the reviewer that if there are large-scale conformational changes induced by the mutation at the interface between DBD and LBD, we may detect by SAXS. However, our size

exclusion (gel filtration) elution profiles (see the figure below) indicate that there is no obvious large-change in size for mutant N407A since the elution peaks (marked by arrows) are nearly identical. This observation suggests that the conformational change induced by the N407A mutation is relatively small or local, which may not be detectable by SAXS since SAXS is not sensitive to such local changes. For this reason, we did not pursue SAXS studies further for this mutant.

To further address whether N407A mutation results in changes at the DBD-LBD interface, we directed our efforts by using the highly sensitive tryptophan fluorescence. We rationalize that because N407 and W200 locate in LBD and DBD separately, a reduction of fluorescence emission of W200 in response to N407A mutation reflects directly the conformational change around W200. Indeed, our data provide evidence of the structural changes of W200 surroundings as a result of the mutation N407A (Fig. 3f). We further clarify and emphasize this finding in page 12.

More detailed and systematic studies on this and other mutations at the LBD-DBD interface are critical to our next-step studies, structurally and functionally. Because hER α protein samples are prone to aggregation, large-scale protein purification for each mutant is required for successful chromatography-coupled SAXS data acquisition. As such, systematic SAXS-related structural studies of these interfacial mutants as well their functional studies will be a key focus of our efforts in the next few years, beyond the scope of this manuscript.

3. In the current manuscript authors proposed a novel L-shaped architecture of hER α , which they think differs from other members of nuclear receptor family (and other ERA structures estimated via SAXS). However, more discussion should be included comparing their model and its functional relevance with the DBD-LBD inter-domain architecture seen in other nuclear receptors.

We now include a new figure (suppl. Figure 15) to illustrate the overall architectures of currently known members of the nuclear receptor family (including USP/EcP as Reviewer #1 suggested). It is striking that both hER α and USP/EcP adopt an extended L-shape (Supp. Figure 15a-b; see also below), although there is a major difference in the residues involved at the interface. Specifically, in the case of USP/EcP, USP LBD directly interact with DNA, while hER α -LBD interact with its DBD. We note that while hER α and USP/EcP bind inverted repeat (IR) DNA (each with a different spacer), the four other structurally known nuclear receptor complexes (PPAR γ -RXR α , RAR β -RXR α , RXR α -LXR β , and HNF-4 α ; Supp. Figure 15c) bind direct repeat (DR) DNA. While these four are similar in all taking a relatively compact assembly, compared to hER α and USP/EcP, there is a major difference in the interfacial residues involved in domain-domain contacts. Namely, RAR β -RXR α uses the loop region between H9 and H10 helices of RAR β -LBD, while RXR α -LXR β and

HNF-4 α have used H9 at the interface. In contrast, hER α relies on the two beta-strands between helices H5 and H6 to make domain-domain contacts (see the nomenclature in Suppl. Figure S11). Of note, a similar beta-strand region is involved in PPAR γ -RXR α 's domain-domain interaction, where Phe347 from the H5-H6 connecting beta-strand region of PPAR γ LBD is shown to functionally mediate the interface (ref. 5), although there is a difference from the solution structure ref. 7). Overall, despite the high similarity of individual LBD and DBD, it is clear that a different domain surface is involved in each domain-domain contact. Finally, we also updated the fitting our hER α ensemble-structure into the 22-Å EM map to further illustrate the functional relevance of our hER α model with regard to its consistency with coactivator binding (see updated Suppl. Figure 11). We have included additional discussion to reflect these points (p. 14).

I would also suggest a good blob model of domain arrangement as final figure would enhance the clarity and understanding of the system to the readers.

As suggested by the reviewer, a new schematic blob-model figure (Fig. 5) is now included in the main text (also shown below), where hER α domains are represented each as a colored blob. To improve the clarity, we use this figure to summarize our major findings. Specifically, Fig. 5a shows the domain arrangements, Fig. 5b depicts a close-up view of the domain interface as an allosteric channel for domain cross-talk, and Fig. 5c schematically illustrates the tryptophan fluorescence that can be used to probe the surrounding changes from structural perturbation. In a proper reference to this new figure, we have updated the main text accordingly (pages 12-14).

REVIEWERS' COMMENTS:

Reviewer #3 (Remarks to the Author):

I think that the overall analysis and description of the revised draft was further improved based on reviewer's comments. The registration to SASBDB is also a wonderful contribution to readers. However, several minor modifications are still necessary.

1. Considering from the pixel size of the CCD detector, we think that the number of measurement data points is very small. Did this analysis perform after binning the data to improve S/N of data? If so, you need to describe it.

2. Guinier Plot was added to figure 2. But the q region where the Guinier approximation was done based on the condition of $q \times R_g < 1.3$ is not shown. You should add that range to Table 2, and correction of this Guinier Plot is also necessary. Please refer to Guinier Plot displayed on SASDDU8 page.

3. The software used in Table 2 is not summarized. In order to register with SASBDB, you would calculate the molecular weight etc. based on the partial specific volume. Such information should also be summarized in Table 2.

(eg.)

SAS data processing (2D -> 1D convert, Background subtraction, etc ...)

Comparison between exp. and theo. profile: Crysol

Calculation of v , $\Delta\rho$ values

etc ...

Please update Table 2 based on the registration information of SASDDU8 and the publication guideline.

4. Since R_g is 38 Å, the data range used for analysis is 9.5 based on $q (= 0.25) \times R_g$. However, in Figure 2d, it appears that the region of $q > 0.15$ does not fit at all. However, as long as I confirm the data registered in SASDDU8, it seems that it fitted as much as possible to the region of $q \sim 0.25$. Is there something wrong with the making and the display in Figure 2d?

Reviewer #4 (Remarks to the Author):

The authors have address our prior concerns and we now feel that that manuscript is suitable for publication in NCOMMS.

Point-by-point response to reviewer's comments

Re: NCOMMS-18-05656A

“Multidomain architecture of estrogen receptor reveals interfacial cross-talk between its DNA-binding and ligand-binding domains”

Reviewer #3 (Remarks to the Author):

I think that the overall analysis and description of the revised draft was further improved based on reviewer's comments. The registration to SASBDB is also a wonderful contribution to readers. However, several minor modifications are still necessary.

1. Considering from the pixel size of the CCD detector, we think that the number of measurement data points is very small. Did this analysis perform after binning the data to improve S/N of data? If so, you need to describe it.

Indeed, we applied the binning of scattering data to improve the S/N ratio. Accordingly, we have added the sentence of "Data reduction resulted in a final one-dimensional $I(q)$ profile with a bin size of $\Delta q \sim 0.004 \text{ \AA}^{-1}$ " in the Methods.

2. Guinier Plot was added to figure 2. But the q region, where the Guinier approximation was done based on the condition of $q \times R_g < 1.3$, is not shown. You should add that range to Table 2, and correction of this Guinier Plot is also necessary. Please refer to Guinier Plot displayed on SASDDU8 page.

The q -range used for the Guinier fitting is now included in Supplementary Table 3 (i.e., the previous Supplementary Table 2), with a $q \times R_g$ value of 1.24. To be consistent with the Guinier Plot displayed on SASDDU8 page, we have updated Figure 2d accordingly.

3. The software used in Table 2 is not summarized. In order to register with SASBDB, you would calculate the molecular weight etc. based on the partial specific volume. Such information should also be summarized in Table 2. (eg.) SAS data processing (2D \rightarrow 1D convert, Background subtraction, etc ...) Comparison between exp. and theo. profile: Crysol Calculation of v , $\Delta\rho$ values, etc ... Please update Table 2 based on the registration information of SASDDU8 and the publication guideline.

All the information required for our SASDDU8 registration into the SASBDB database are now included in Supplementary Table 3.

4. Since R_g is 38 Å, the data range used for analysis is 9.5 based on $q (= 0.25) \times R_g$. However, in Figure 2d, it appears that the region of $q > 0.15$ does not fit at all. However, as long as I confirm the data registered in SASDDU8, it seems that it fitted as much as possible to the region of $q \sim 0.25$. Is there something wrong with the making and the display in Figure 2d?

We have edited the figure caption of Figure 2d to improve the clarity by adding the sentence of "theoretical SAXS data were the ensemble average of the set of hER α structures above." This should clarify the difference between Figure 2d and the SASDDU8 page, where the fitting plot on the SASDDU8 page was based on individual structures, while Figure 2d was plotted using the ensemble average of a set of structures.